# Environmental and Health-Related Lifecycle Impact Assessment of Reduced-Salt Meals in Japan

## Keiji Nakamura [1,2,*] and Norihiro Itsubo [3]

1   Research Center for Sustainability Science of Advanced Research Laboratories, Tokyo City University, Yokohama 224-8551, Japan
2   Research and Business Planning Department, Ajinomoto Co., Inc., Tokyo 104-8315, Japan
3   Faculty of Environmental Studies, Tokyo City University, Tokyo 158-8557, Japan; itsubo-n@tcu.ac.jp
*   Correspondence: keijin@tcu.ac.jp; Tel.: +81-45-910-0104

**Abstract:** To achieve sustainable development goals, meals should promote environmental protection and good health. The excessive salt intake of the Japanese people is one cause of lifestyle diseases. This study evaluated the impact of reducing salt intake on the environment and human health. Over one week, this study compared the lifecycle of a model meal based on a Japanese person's average food intake with a reduced-salt meal, by replacing seasoning/salt with low-salt substitutes. We conducted an inventory assessment of the carbon and water consumption footprints based on the items used in the ingredient and cooking stages. The impact on climate and water consumption was determined using the results of the inventory assessment of the damage factors. We took the global burden of disease result as the health impact of salt intake. The decreased health impact of reduced salt was based on the results of a previous study. The health impact of the ingredient stage of both meals was almost the same. Regarding the assessment of the health impact, the value of the reduced-salt meal was 30% lower than that of the model meal because the salt intake was reduced by 2.8 g per day. We found that the reduced-salt meal could decrease the overall human health impact by 20% because of the reduced incidence of salt-intake-related diseases, despite a small increase in the health impact of reduced-salt seasonings.

**Keywords:** reduced-salt meal; lifecycle assessment; environment; carbon footprint; water footprint; health-related impact assessment; disability-adjusted life years

## 1. Introduction

In 2001, the average life expectancy for Japanese men was 78 years and that for Japanese women was 85 years, whereas in 2010, it was 80 years for men and 86 years for women [1]. In 2001, healthy life expectancy was about 69 years for men and 73 years for women, whereas in 2010, it was 70 years for men and 74 years for women [1]. The difference between healthy life expectancy and average life expectancy varied over the 10 years.

Imbalances in diet and nutrient intake are among the issues causing lifestyle diseases and harming the health of the Japanese people. The Ministry of Health, Labor and Welfare, in its Dietary Reference Intakes for Japanese People, suggests that excessive salt intake is one of the causes of lifestyle diseases (high blood pressure, hyperlipemia, diabetes, and chronic kidney disease) [2]. In terms of the global burden of disease (GBD), the greatest of the 15 dietary risk factors for Japanese people is high sodium [3]. The current average salt intake of Japanese people is about 10 g per person per day (sodium base 3.94 g), which differs from the 8 g per person per day (sodium base 3.15 g) target set out in Health Japan 21 [4] and the 5 g per person per day (sodium base 1.97 g) target of the World Health Organization (WHO) [5].

The worldwide agriculture and livestock industry emits 17.3 billion tons of $CO_2e$ as greenhouse gases (GHGs), which accounts for one-third of the total. The livestock industry

accounts for 57% of global GHGs in the world's food systems, and agriculture for food accounts for 29%; the remaining 14% is from other agricultural products such as cotton and rubber [6]. The food and drink, agriculture, forestry, and fishing industries in Japan emit 70 million tons of $CO_2e$ as GHGs, accounting for approximately 6% of Japan's total GHG emissions [7]. In order to mitigate climate change, GHG emissions from food, which is essential for human life, must also be reduced. The agriculture and livestock industries consume large quantities of water, and in Japan they account for 65% of the total water consumption [8]. Therefore, the environmental impact of food should also include an assessment of water consumption.

To achieve the Sustainable Development Goals (SDGs), we must consider the balance between environmental protection (SDG 13, climatic action; SDG 14, life below water; SDG 15, life on land) and good health and well-being (SDG 3). As an alternative to regular seasonings, reduced-salt seasonings are produced using desalting or desalination treatments or by adding umami substances to improve palatability. Therefore, there is a concern that the carbon footprint of reduced-salt seasonings is higher than that of regular seasonings. It is important to evaluate the environmental and health impacts related to diet to achieve a balance between environmental protection regarding the SDGs and health.

In a study on the environmental and health effects of diet, Guobao et al. [9] proposed the reduction of GHG emissions by decreasing meat consumption, based on the results of health and nutrition surveys, but did not assess the health impact. Springmann et al. [10] evaluated monetary value; the health impact of reduced mortality from heart disease, stroke, cancer and diabetes; and the environmental impact of reduced GHG emissions arising from decreased red meat consumption and/or the increased intake of fruits and vegetables, but did not evaluate salt intake. Stylianou et al. [11] developed the Health Nutritional Index to quantify marginal health effects by combining nutritional health-based and 18 environmental indicators in the US diet, but did not evaluate this in the Japan diet. Nomura et al. [12] established a relationship between salt intake and stomach cancer, cardiovascular disease, and chronic kidney disease in humans by studying disability-adjusted life years (DALYs), but did not evaluate the environmental effects.

This study evaluated the impact of reducing salt on the environment (climate change and water consumption) and human health (DALYs), comparing a model meal based on the average food intake of Japanese people with a reduced-salt version of the model meal. The reduced-salt meal was prepared with the reduced-salt scenario, in which seasonings were changed from those of a model meal to those of the reduced-salt seasonings by replacing seasoning/salt with low-salt substitutes. Although the Japanese-style meal is recognized as having cultural heritage status by the United Nations Educational, Scientific and Cultural Organization (UNESCO, Paris, France) [13], in recent years Japanese people have tended to eat Western-style meals more often [14]. To evaluate the current meals eaten by Japanese people, we classified them into Japanese and Western styles. Because the results of the National Nutrition Survey also suggested that the lack of breakfast increases the risk of nutrient bias [15], this study also evaluated and discussed meals by classifying them into breakfast, lunch, and dinner.

## 2. Materials and Methods

This study assessed the environmental and health-related effects of meals consumed by Japanese people with different levels of salt. The salt intake levels were created by replacing seasoning/salt with low-salt substitutes. We conducted an inventory assessment of the carbon and water consumption footprints based on the items used in the ingredient and cooking stages. The impact on climate and water consumption was determined using the results of the inventory assessment of the damage factors. We took the GBD as the health impact of salt intake. The decreased health impact of reduced salt was based on the results of a previous study. A comprehensive impact assessment was conducted by adding the effects on climate and water and the health impact of salt intake. Figure 1 shows the structure of this study.

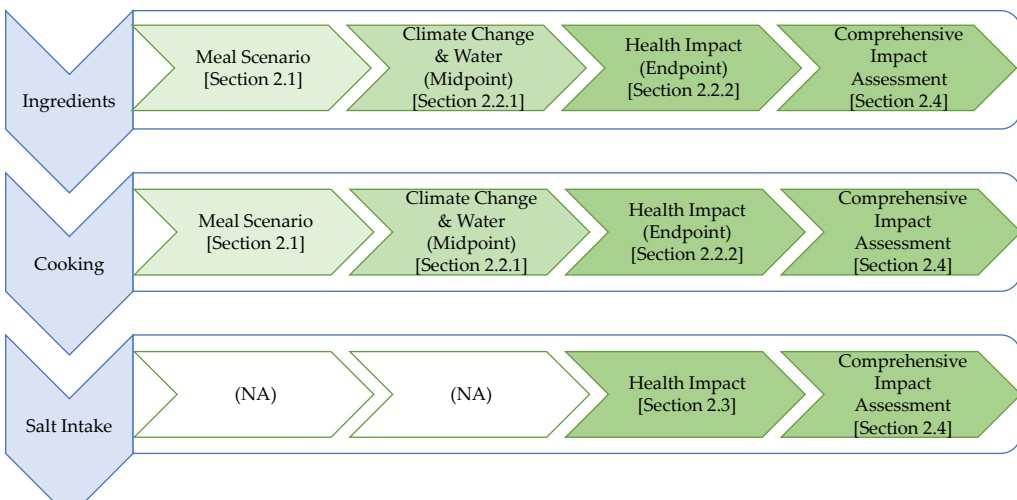

**Figure 1.** Structure of this study.

## 2.1. Preparation of a Comparison Meal Scenario

A one-week model meal list using a standard scenario and a reduced-salt meal list using a reduced-salt scenario were prepared for comparison.

The standard scenario was based on the average daily food intake of people aged 20 and over (the sum of breakfast, lunch, dinner and snacks, 1930 kcal), as shown in the food intake by food group in the National Health and Nutrition Survey of 2018 [16]. The daily average intake of those aged 20 years and over was multiplied by seven (13,510 kcal) to provide the one-week model meal list. In preparing this list, we combined one week's total volume of main ingredients (seasoning/salt, fish, chicken, pork, beef, rice, and wheat bread) and one dish to replicate the balance breakdown of the base data according to data on the energy and intake of main ingredients in the 2018 National Health and Nutrition Survey: breakfast, 434.3 kcal per day, 3040.1 kcal per week; lunch, 580.0 kcal per day, 4060.0 kcal per week; dinner, 785.0 kcal per day, 5495.0 kcal per week; and snacks, 130.7 kcal per day, 914.9 kcal per week [17]. One dish was selected from the dishes published on the websites AJINOMOTO PARK [18], Lettuce Club [19], and Japanese Confectionery Fukuya [20], and the selection method was informed by consultation with a Japanese nutrition specialist. The energy for the model meal list was 1893 kcal, which was 2% lower than that for the daily average intake of people aged 20 years and over. We considered that the model meal list fell within a reasonable margin.

The reduced-salt meal list was prepared with the reduced-salt scenario, in which seasonings were changed from those of the standard scenario to those of the reduced-salt food list prepared by the academic Japanese Society of Hypertension [21]. Specifically, in the basic seasonings, salt was changed to reduced-salt seasoning, with a salt reduction of 50% achieved by adding potassium chloride; soy sauce was changed to reduced-salt soy sauce, with a salt reduction of 50% achieved by desalting; and miso was changed to reduced-salt miso, with a salt reduction of 25% achieved by changing the composition. Flavorings were changed to reduced-salt flavouring for Japanese bouillon, with a reduction of 60%, reduced-salt flavouring for Western bouillon, with a reduction of 40%, and reduced-salt flavouring for Chinese bouillon, with a reduction of 40%. These reductions in salt were achieved through changes in composition.

Table 1 shows the one-week model meal list prepared based on food intake by food groups [16] and the energy and intake of the main ingredients [17] in the 2018 National Health and Nutrition Survey. This study used a Japanese-style meal in which the main ingredients were salted dried fish, raw fish and tofu, and cooking methods were hot pot, simmering, and fried rice. We used a Western-style meal with ingredients of sandwiches, gratin, spaghetti and curry, and cooking methods from overseas.

**Table 1.** One-week model meal list (values in parentheses indicate salt intake for the reduced-salt meal).

| | Day 1 | Day 2 | Day 3 | Day 4 | Day 5 | Day 6 | Day 7 |
|---|---|---|---|---|---|---|---|
| 13,253 kcal per week<br>65.4 g salt per week (45.8) | 1862 kcal per day<br>14.6 g salt per day (9.8) | 1817 kcal per day<br>9.4 g salt per day (5.6) | 1918 kcal per day<br>9.4 g salt per day (6.6) | 1981 kcal per day<br>5.4 g salt per day (3.9) | 1851 kcal per day<br>8.4 g salt per day (5.8) | 1960 kcal per day<br>9.3 g salt per day (7.5) | 1864 kcal per day<br>9.2 g salt per day (6.7) |
| Breakfast | B-1<br>Rice<br>Horse mackerel<br>Mushroom miso soup<br>Fermented soybean<br>Oolong tea<br><br>Japanese<br>Fish = 56 g<br>Rice = 130 g<br>479 kcal<br>3.9 g salt (3.2) | B-2<br>Rice omelette (half)<br>Milk<br>Coffee<br><br>Western<br>Chicken = 30 g<br>Rice = 135 g<br>442 kcal<br>1.0 g salt (0.8) | B-3<br>Sandwich with cabbage,<br>ham, cheese<br>Strawberry<br>Milk<br>Coffee<br>Western<br>Ham = 30 g<br>Bread = 60 g<br>416 kcal<br>2.2 g salt (2.2) | B-4<br>Sandwich with zucchini<br>Coffee<br><br><br>Western<br>None<br>Bread = 45 g<br>377 kcal<br>1.7 g salt (1.6) | B-5<br>Gratin with broccoli and<br>egg<br>Oolong tea<br>Coffee<br><br>Western<br>Ham = 30 g<br>None<br>552 kcal<br>3.1 g salt (2.5) | B-6<br>Rice<br>Tofu and Sichuan<br>vegetables<br>Banana<br>Oolong tea<br>Japanese<br>None<br>Rice = 130 g<br>365 kcal<br>3.0 g salt (2.3) | B-7<br>Rice<br>Hot and sour soup<br>Oolong tea<br><br>Japanese<br>Pork = 20 g<br>Rice = 130 g<br>312 kcal<br>3.4 g salt (1.8) |
| Lunch | L-1<br>Wheat noodles with fish,<br>sausage and vegetables<br>Potato soup<br>Oolong tea<br><br>Japanese<br>Fish = 20 g<br>Wheat noodle = 250 g<br>544 kcal<br>5.8 g salt (3.8) | L-2<br>Fried rice with egg and<br>green onion<br>Oolong tea<br><br>Japanese<br>None<br>Rice = 200 g<br>551 kcal<br>3.2 g salt (1.8) | L-3<br>Bread<br>Creamed chicken stew<br>Oolong tea<br><br>Western<br>Chicken = 56 g<br>Bread = 60 g<br>569 kcal<br>3.5 g salt (2.3) | L-4<br>Rice<br>Quiche with bacon and<br>spinach<br>Apple<br>Oolong tea<br>Western<br>Ham = 27 g<br>Rice = 100 g<br>799 kcal<br>1.5 g salt (1.2) | L-5<br>Japanese-style salad with<br>dumpling<br>Oolong tea<br><br>Japanese<br>Pork = 38 g<br>Wheat = 30 g<br>370 kcal<br>2.1 g salt (1.0) | L-6<br>Rice mixed with seasoned<br>vegetables<br>Oolong tea<br><br>Japanese<br>Beef = 50 g<br>Rice = 150 g<br>513 kcal<br>2.9 g salt (1.8) | L-7<br>Sandwich with tuna and<br>carrot<br>Oolong tea<br><br>Western<br>Fish = 70 g<br>Bread = 120 g<br>728 kcal<br>3.5 g salt (3.2) |
| Dinner | D-1<br>Rice<br>Hot pot with pork back<br>ribs and cabbage<br>Fermented soybeans<br>Wine<br>Oolong tea<br>Japanese<br>Pork = 50 g<br>Rice = 150 g<br>676 kcal<br>4.9 g salt (2.8) | D-2<br>Rice<br>Pieces of raw fish<br>Tofu miso soup<br>Wine<br>Oolong tea<br><br>Japanese<br>Fish = 222 g<br>Rice = 150 g<br>760 kcal<br>5.1 g salt (3.0) | D-3<br>Rice<br>Yellowtail and root<br>vegetables<br>Japanese radish salad<br>Fermented soybeans<br>Wine<br>Oolong tea<br>Japanese<br>Fish = 100 g<br>Rice = 100 g<br>869 kcal<br>3.6 g salt (2.1) | D-4<br>Rice<br>Roast beef<br>Coke<br><br><br>Western<br>Beef = 150 g<br>Rice = 150 g<br>741 kcal<br>2.2 g salt (1.1) | D-5<br>Rice<br>Chicken ham<br>Fermented soybeans<br>Pineapple<br>Coke<br><br>Japanese<br>Chicken = 150 g<br>Rice = 150 g<br>865 kcal<br>3.2 g salt (2.2) | D-6<br>Curry and rice with pork<br>Boiled egg<br>Coke<br>Watermelon<br><br>Western<br>Pork = 75 g<br>Rice = 200 g<br>993 kcal<br>3.4 g salt (3.4) | D-7<br>Spaghetti with tomato and<br>sausage<br>Beer<br>Oolong tea<br>Nashi pear<br><br>Western<br>Ham = 34 g<br>Pasta = 165 g<br>736 kcal<br>2.3 g salt (1.7) |
| Snack | Baked sweet potato<br><br>163 kcal<br>0 g salt | Yogurt<br><br>64 kcal<br>0 g salt | Yogurt<br><br>64 kcal<br>0 g salt | Yogurt<br><br>64 kcal<br>0 g salt | Yogurt<br><br>64 kcal<br>0 g salt | Arrowroot cake and<br>orange<br>89 kcal<br>0 g salt | Coffee jelly and grapefruit<br><br>88 kcal<br>0 g salt |

*2.2. Environmental Impact Assessment*

A lifecycle assessment for environmental impact was conducted according to the following procedure. The item input for ingredients was selected from the one-week model meal and reduced-salt meal lists as shown Table 1. The item input for cooking was then applied, which detailed the gas and electric power consumption of the cooking method required for each dish, and consumption was calculated in accordance with a previous study by Tsuda et al. [22].

### 2.2.1. Midpoint Assessment

We conducted an inventory assessment using a Japanese lifecycle inventory database, IDEA (version2.3, Sustainable Management Promotion Organization (SuMPO), Tokyo, Japan) [23], which comprehensively covers nearly all economic activities of Japanese businesses, to determine the carbon footprint (CFP) and water consumption footprint (WCFP) related to the environmental aspects of food, based on the item input for the ingredient and cooking stages. We used greenhouse gas emission factor as $CO_2$ intensity and water consumption factor as water intensity on IDEA v2.3. These factors were the item's total $CO_2$ equivalent emissions from cradle to gate, i.e., from nursery plants to cultivation using water and fertilizer on agricultural farms.

$$CFP = \sum_{s} \sum_{i} \{(Amount\ of\ the\ item\ input)_{i,s} \times (CO_2\ intensity)_{i,s}\} \tag{1}$$

$$WCFP = \sum_{s} \sum_{i} \{(Amount\ of\ the\ item\ input)_{i,s} \\ \times (Water\ consumption\ intensity)_{i,s}\} \tag{2}$$

Here, $i$ is the item input and $s$ is the lifecycle stage. The calculation example is shown in Table A1 in Appendix A.

The primary data obtained for the inventory assessment of reduced-salt seasonings, i.e., the input ratios, are shown in Table A2 in Appendix A. The data for the reduced-salt soy sauce, for which no primary data were obtained, were calculated based on the following procedures with reference to the basic method of production in Kikkoman's reduced-salt soy sauce book [24]. Normal soy sauce was desalted using an ion-exchange membrane, which applied the salt-removal technology used to process seawater. Salt Remover S3 was operated for 90 min to make 1 L of reduced-salt soy sauce (500 mL per 45 min, two batches) [25]. The inventory data for 1 L of reduced-salt soy sauce were calculated by multiplying the IDEA electricity inventory data (average of 10 general electricity utilities, FY2012, 1 kWh) of 0.75 kW; these prepared data were added to the IDEA soy sauce inventory data. The reduced-salt miso recipe, for which no primary data were available because the production method is not disclosed by the manufacturer, was calculated using the following procedure. We calculated the average multiplication factor between regular and reduced-salt seasoning in terms of the carbon and water inventory. The inventory data for reduced-salt miso were prepared by multiplying the IDEA miso inventory data by the average multiplication factor.

### 2.2.2. Endpoint Assessment

The health impact in terms of climate and water was determined using the results of the inventory assessment of damage factors, SSP2 from Tang et al. for climate change [26]

and CFagr_agri users of Japan in Global Guidance for Life Cycle Impact Assessment Indicators (GLAM) volume 2 for annual average water consumption [27]:

$$
\begin{aligned}
Health\ Impact_{climate\ \&\ water} \\
= \sum_s \sum_i (CFP_{i,s} \times Damage\ Factor_{climate}) \\
+ \sum_s \sum_i (WCFP_{i,s} \times Damange\ Factor_{water})
\end{aligned}
\tag{3}
$$

Here, *i* is the item input and *s* is the lifecycle stage.

### 2.3. Health Impact Assessment

#### 2.3.1. Health Impact of Salt Intake

We used the GBD result as the health impact of salt intake. The health impact of salt intake for the one-week model meal list with the standard scenario was 603,211 DALYs per Japanese population per year, as the high-sodium diet of total all-cause GBD divided by 126,150,000, the Japanese population in 2017 and 365 days; $1.31 \times 10^{-5}$ DALYs per person per day represents the base data for the impact of excessive salt intake on health disorders. This impact was considered to be caused by salt intake according to the one-week model meal list with the standard scenario as $1.31 \times 10^{-5}$ DALYs per person per day divided by 9.3 g per day = $1.40 \times 10^{-6}$ DALYs per gram; this represents the base data for the impact of 1 g salt intake on health. The model for the health impact of salt was calculated as follows: health impact of 1 g of salt intake multiplied by salt intake for one meal of the one-week model meal list.

#### 2.3.2. Decreased Health Impact of Reduced Salt

The decreased health impact of reduced salt was based on the results of a previous study by Nomura et al. [11]. In that study, they simulated DALYs per 100,000 people for three diseases (stomach cancer, cardiovascular disease, and chronic kidney disease) using multiple scenarios with different forecast levels of salt intake with the autoregressive integrated moving average model. We found that DALYs per 100,000 people for the three diseases with the lowest and highest values in the best scenario (achievement of the WHO's goal of 5 g per person per day in 2040) and worst scenario (continued intake of 9.9 g per person per day) followed normal distributions of standard deviations. We generated 1 million normally distributed random numbers (Monte Carlo simulation) between the lowest and highest values in 2040 and averaged the random numbers for each scenario. This confirmed that there was a quantitative advantage in the difference in the mean of random numbers in 2040 for the two scenarios regarding the three diseases (stomach cancer, cardiovascular disease, and chronic kidney disease), and the total 244.7 DALYs per 100,000 persons per year was divided by 4.9 g per day of the reduced-salt intake difference and 100,000 persons and 365 days, and a decreased health impact index for reduced salt of $1.37 \times 10^{-6}$ DALYs per gram was obtained, as shown in Table 2.

$$
\begin{aligned}
\Delta 1g\ Reduced\ Salt\ Health\ Impact \\
= (\Delta Stomach\ Cancer + \Delta Cardiovascolar\ Disease \\
+ \Delta Chronic\ Kidney\ Disease) \\
\div \Delta Salt\ Intake \div 100,000\ persons \div 365 days
\end{aligned}
\tag{4}
$$

The health impact of the reduced-salt meal was obtained from the difference between the impact of the salt model and reduced salt intake per meal in grams multiplied by the decreasing health impact index by reduced salt:

$$
\begin{aligned}
Health\ Impact_{salt\ reduced} \\
= Health\ Impact_{salt\ model} \\
- (\Delta Reduced\ Salt\ Intake \\
\times \Delta 1g\ Reduced\ Salt\ Health\ Impact)
\end{aligned}
\tag{5}
$$

In concert with Equation (5) for the B-2 Rice omelette (half) in Table 1, the health impact of 1 g of salt intake = $1.40 \times 10^{-6}$ DALYs per gram was multiplied by the salt intake of B-2 = 1.0 g, the difference between the impact of the salt model and reduced salt intake per meal in 0.2 g was multiplied by the decreasing health impact index = $1.37 \times 10^{-6}$ DALYs per gram; the health impact of the reduced-salt meal of B-2 = $1.126 \times 10^{-6}$ DALYs was obtained by the former value minus the later value.

**Table 2.** Mean values based on DALY results from [11] for salt intake and three related diseases.

| 2040 | Best Scenario | Worst Scenario | Difference | Total |
|---|---|---|---|---|
| Forecast salt intake (grams) (SI) | 5.0 | 9.9 | 4.9 | Not applicable |
| Stomach cancer (SC) | 465.9 | 530.3 | 64.4 | |
| Cardiovascular disease (CVD) | 3886.5 | 4040.0 | 153.5 | 244.7 |
| Chronic kidney disease (CKD) | 612.9 | 639.7 | 26.8 | |

### 2.4. Comprehensive Impact Assessment

Regarding the one-week model meal and reduced-salt meal lists, the endpoint assessment results for DALYs at the ingredient and cooking stages were added to the health-impact DALYs for salt or reduced salt, and a comprehensive impact assessment was conducted:

$$\begin{aligned}
&Health\ Impact_{environment\ \&\ nutiient} \\
&= Health\ Impact_{climate\ \&\ water} + Health\ Impact_{salt}
\end{aligned} \tag{6}$$

## 3. Results and Discussion

### 3.1. Results

Our results are presented as follows: climate change and water consumption footprints of the midpoint assessment are presented in Section 3.1.1, the health impact of salt intake is discussed in Section 3.1.2, and the comprehensive impact is discussed in Section 3.1.3. The health effects of Japanese and Western meal styles are compared in Section 3.2.1, and their respective environmental impacts at the endpoint assessment are compared in Section 3.2.2. Figure 2 shows the structure of this section.

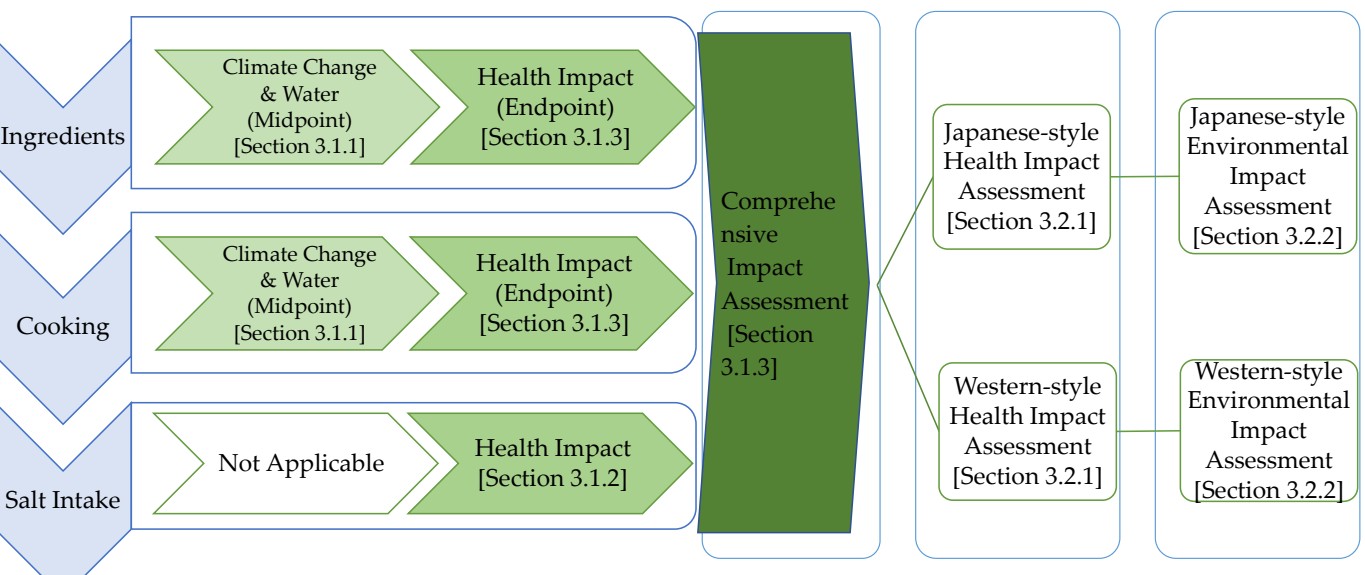

**Figure 2.** Structure of this section.

3.1.1. Results of the Environmental Impact Assessment

Table 3 shows the environmental impact assessments for climate change (CC; kg $CO_2e$) and water consumption (WC; $m^3$) of the one-week model meal and reduced-salt meal lists.

**Table 3.** Environmental impact assessments for climate change (CC; kg $CO_2e$) and water consumption (WC; $m^3$) for the one-week model meal and reduced-salt meal lists.

| | | One Week | | One Day | |
|---|---|---|---|---|---|
| | | Model | Reduced Salt | Model | Reduced Salt |
| Climate change (kg $CO_2e$) | Ingredients | 21.6 | 21.8 | 3.09 | 3.11 |
| | Cooking | 1.75 | 1.75 | 0.25 | 0.25 |
| | Total | 23.4 | 23.5 | 3.34 | 3.36 |
| Water consumption ($m^3$) | Ingredients | 3.24 | 3.25 | 0.46 | 0.47 |
| | Cooking | $1.12 \times 10^{-4}$ | $1.12 \times 10^{-4}$ | $1.60 \times 10^{-5}$ | $1.60 \times 10^{-5}$ |
| | Total | 3.24 | 3.25 | 0.46 | 0.47 |

The CC values of the ingredient stage of the model meal and the reduced-salt meal were 21.6 and 21.8 kg $CO_2e$ per week, respectively. The latter value was 1% higher than the former value with reduced-salt seasoning desalination and additional umami substances for improving palatability. The CC value of the cooking stage of both meals was 1.75 kg $CO_2e$ per week, less than 10% of the ingredient stage, and we found that the cooking stage was not significant in the CC for meals. Table 4 shows the CC comparing meal style (Japanese, Western). The average CC value of the Japanese-style model meal was 0.83 kg $CO_2e$ per meal in a range of 0.44~1.49 kg $CO_2e$ per meal. The average CC value of the Japanese-style reduced-salt meal was 0.85 kg $CO_2e$ per meal in a range of 0.45~1.50 kg $CO_2e$ per meal, 1% higher with reduced-salt seasoning than the CC of the Japanese-style model meal. The CC value of both the Western-style meal model and the reduced-salt meal was an average of 1.28 kg $CO_2e$ per meal in a range of 0.54~3.68 kg $CO_2e$ per meal. The value of Western-style meals was about 2.5 times higher at maximum and about 1.5 times higher on average than the value of the Japanese-style meals because of the large amount of beef and other meats. Table A3 in Appendix A shows the CC values comparing breakfast, lunch, and dinner. The CC value for each meal with the reduced-salt seasoning was 1% higher compared to the model meals.

**Table 4.** Climate change (CC), comparing meal styles (Japanese and Western).

| Climate Change kg-$CO_2$ Per Meal | Japanese (Model) | Japanese (Reduced-Salt) | Western (Model) | Western (Reduced-Salt) |
|---|---|---|---|---|
| Maximum data | 1.49 | 1.50 | 3.68 | 3.68 |
| Mean data | 0.83 | 0.85 | 1.28 | 1.28 |
| Median data | 0.67 | 0.68 | 1.03 | 1.04 |
| Minimum data | 0.44 | 0.45 | 0.54 | 0.54 |

The WC values of the ingredient stage of the model meal and the reduced-salt meal were 3.24 and 3.25 $m^3$ per week, respectively. The latter value was 0.3% higher than the former value with reduced-salt seasoning. The WC of the cooking stage of both meals was $1.12 \times 10^{-4}$ $m^3$ per week, which was less than 0.01% of the ingredient stage, and it was found that the cooking stage was not significant in the WC for meals. Table 5 shows the WC comparing meal styles (Japanese, Western). The WC of both the Japanese-style meal and the reduced-salt meal was an average of 0.16 $m^3$ per meal in a range of 0.04~0.22 $m^3$ per meal. The WC of both the Western meal and the reduced-salt meal was an average of 0.14 $m^3$ per meal in a range of 0.05~0.36 $m^3$ per meal. Although the maximum and minimum values of the Japanese-style meal were smaller than the values of the Western-

style meal, the average value increased by about 10%, owing to the higher rice intake. Table A4 in Appendix A shows the WC values comparing breakfast, lunch, and dinner. The WC value for the lunch with reduced-salt seasoning was 1% higher compared to the model lunch. The WC value for the dinner with reduced-salt seasoning was 0.5% higher compared to the model dinner.

**Table 5.** Water consumption (WC), comparing meal styles (Japanese and Western).

| Water Consumption m³ Per Meal | Japanese (Model) | Japanese (Reduced-Salt) | Western (Model) | Western (Reduced-Salt) |
|---|---|---|---|---|
| Maximum data | 0.22 | 0.22 | 0.36 | 0.36 |
| Mean data | 0.16 | 0.16 | 0.14 | 0.14 |
| Median data | 0.15 | 0.15 | 0.10 | 0.10 |
| Minimum data | 0.04 | 0.04 | 0.05 | 0.05 |

### 3.1.2. Results of the Health Impact Assessment

Table 6 shows the variable salt intake (SI) and reduction in health impact_salt (HI_s) values for the one-week model meal and reduced-salt meal lists. Table 7 shows SI comparing meal styles (Japanese and Western). The average SI values for the Japanese and Western meals were 3.7 and 2.4 g per day, respectively. The value was higher for the Japanese meal than for the Western meal; however, the reduction for the Japanese meal was 2.3 g per day, which was higher than that for the Western meal. The reason for the high salt intake with the Japanese meal was that many dishes use high-salt seasoning such as miso and soy sauce, and much salt is used for curing vegetables and seaweed, such as pickled vegetables and fish boiled in soy sauce [28].

**Table 6.** Values of variables salt intake (SI) and reduction in health impact_salt (HI_s) for one-week model meal and reduced-salt meal lists.

| | Salt Intake (SI) | | Health Impact_salt (HI_s) | |
|---|---|---|---|---|
| | Grams Per Week | Grams Per Day | DALYs Per Week | DALYs Per Day |
| Model | 65.4 | 9.3 | $9.13 \times 10^{-5}$ | $1.30 \times 10^{-5}$ |
| Reduced salt | 45.8 | 6.5 | $6.36 \times 10^{-5}$ | $0.91 \times 10^{-5}$ |
| Difference | 19.6 | 2.8 | $2.77 \times 10^{-5}$ | $0.39 \times 10^{-5}$ |

**Table 7.** Salt intake (SI) for meal styles (Japanese and Western).

| Salt Intake g Salt Per Meal | Japanese (Model) | Japanese (Reduced-Salt) | Western (Model) | Western (Reduced-Salt) |
|---|---|---|---|---|
| Maximum data | 5.8 | 3.2 | 3.5 | 3.4 |
| Mean data | 3.7 | 2.3 | 2.4 | 2.0 |
| Median data | 3.4 | 2.2 | 2.2 | 1.9 |
| Minimum data | 2.1 | 1.0 | 1.0 | 0.8 |

Table A5 in Appendix A shows the SI for breakfast, lunch, and dinner. The average SI values for the Japanese breakfast, lunch, and dinner were 2.6, 3.2, and 3.5 g, respectively. The average SI values for Western meals were 2.1, 2.1, and 2.3 g per meal, respectively. Since the average SI for dinner was high, reducing salt intake at dinner is arguably likely to be more effective than reducing it at other meals.

A breakdown of the sources of salt for Japanese people is as follows: seasonings, 45% (salt, soy sauce, ketchup, mayonnaise, and miso); flavourings, 31% (Japanese bouillon, Western bouillon, curry powder, Chinese paste, and Chinese bouillon); and processed foods, 24% (bread, wheat noodles, dried fish, fermented soybeans, ham, sausages, and cheese). The usage ratios of flavourings, which are products with seasonings such as salt and soy

sauce, are increasing. The amount of salt added through flavourings during cooking is difficult to determine. However, in this study, flavourings were also examined, and it was suggested that the use of reduced-salt flavourings could reliably reduce salt regardless of the meal style.

### 3.1.3. Results of the Comprehensive Impact Assessment

Table 8 shows the results of the comprehensive impact assessment (CIA). The one-day CIAs of the ingredient stages of the model meal and the reduced-salt meal were $0.675 \times 10^{-5}$ and $0.679 \times 10^{-5}$ DALYs per day, respectively. The value of the one-day CIA of ingredients was 0.6% higher for the reduced-salt meal than for the model meal. The one-day CIAs with different levels of salt intake of the model meal and the reduced-salt meal were $1.30 \times 10^{-5}$ and $0.91 \times 10^{-5}$ DALYs per day, respectively. The value of the one-day CIA for different levels of salt intake was 30% lower for the reduced-salt meal than the model meal because salt intake was reduced by 2.8 g per day. The one-day CIAs of the model meal and the reduced-salt meal were $2.013 \times 10^{-5}$ and $1.627 \times 10^{-5}$ DALYs per day, respectively. The total value of the one-day CIA was 20% lower for the reduced-salt meal than the model meal. We found that the reduced-salt meal would have a 20% lower overall human health impact because of the reduced incidence of salt-intake-related diseases, despite the small increase in the health impact of reduced-salt seasonings.

**Table 8.** Results (DALYs) of the comprehensive impact assessment (CIA).

| (DALYs) | One-Week Model | One-Week Reduced Salt | One-Day Model | One-Day Reduced Salt |
|---|---|---|---|---|
| Ingredients | $4.73 \times 10^{-5}$ | $4.76 \times 10^{-5}$ | $0.675 \times 10^{-5}$ | $0.679 \times 10^{-5}$ |
| Cooking | $0.26 \times 10^{-5}$ | $0.26 \times 10^{-5}$ | $0.038 \times 10^{-5}$ | $0.038 \times 10^{-5}$ |
| Salt | $9.13 \times 10^{-5}$ | $6.36 \times 10^{-5}$ | $1.300 \times 10^{-5}$ | $0.910 \times 10^{-5}$ |
| Total | $1.41 \times 10^{-4}$ | $1.14 \times 10^{-4}$ | $2.013 \times 10^{-5}$ | $1.627 \times 10^{-5}$ |

As a limitation, this study assessed the environmental and health-related effects of meals consumed by Japanese people with different levels of salt. The salt intake levels were created by replacing seasoning/salt with low-salt substitutes. In order to reduce the salt intake of Japanese people, it is important not only to replace ingredients with low-salt substitutes but also to reduce added salt, and educational interventions are very important.

This study only covered the data of 5743 Japanese people, without including other countries. The results only evaluated average intake, not that according to sex and age. This study only assessed salt intake, which is the most problematic nutrient for Japanese people. We hope that other nutrients, including the other 14 top GBD nutrients in order of priority, will be assessed comprehensively in terms of their environmental and health impacts by sex and age.

### 3.2. Discussion

### 3.2.1. Comparison of Meal Styles (Japanese and Western) on Health Impact Assessment

Tables 9–11 show the results of the health impact (HI) comparison of meal styles (Japanese and Western) and ingredients, cooking, and salt intake stages for the model and the reduced-salt meals. The HI value of the ingredient stage of the Western meal was 25% higher than that of the Japanese model meal, equal to a ten-fold difference, because it contained a large amount of beef and other meats. The SI was 50% higher for the Japanese meal than the Western meal, which agreed with the analysis presented in Table 7. The amount of salt reduction for the Japanese meal was high; the HI of the salt intake for the reduced-salt Japanese meal was $3.31 \times 10^{-6}$ DALYs per meal, nearly the same as that for the Western meal.

**Table 9.** Results of the health impact (HI) of ingredient comparison of meal styles (Japanese and Western).

| HI of Ingredient DALYs Per Meal | Japanese (Model) | Japanese (Reduced-Salt) | Western (Model) | Western (Reduced-Salt) |
|---|---|---|---|---|
| Maximum data | $3.22 \times 10^{-6}$ | $3.23 \times 10^{-6}$ | $3.41 \times 10^{-6}$ | $3.41 \times 10^{-6}$ |
| Mean data | $1.92 \times 10^{-6}$ | $1.95 \times 10^{-6}$ | $2.43 \times 10^{-6}$ | $2.44 \times 10^{-6}$ |
| Median data | $1.77 \times 10^{-6}$ | $1.78 \times 10^{-6}$ | $1.85 \times 10^{-6}$ | $1.86 \times 10^{-6}$ |
| Minimum data | $1.08 \times 10^{-6}$ | $1.09 \times 10^{-6}$ | $0.94 \times 10^{-6}$ | $0.95 \times 10^{-6}$ |

**Table 10.** Results of the health impact (HI) of cooking comparisons of meal styles (Japanese and Western).

| HI of Cooking DALYs Per Meal | Japanese (Model) | Japanese (Reduced-Salt) | Western (Model) | Western (Reduced-Salt) |
|---|---|---|---|---|
| Maximum data | $8.38 \times 10^{-8}$ | $8.38 \times 10^{-8}$ | $4.48 \times 10^{-7}$ | $4.48 \times 10^{-7}$ |
| Mean data | $5.83 \times 10^{-8}$ | $5.83 \times 10^{-8}$ | $1.42 \times 10^{-7}$ | $1.42 \times 10^{-7}$ |
| Median data | $6.48 \times 10^{-8}$ | $6.48 \times 10^{-8}$ | $1.01 \times 10^{-7}$ | $1.01 \times 10^{-7}$ |
| Minimum data | $1.10 \times 10^{-8}$ | $1.10 \times 10^{-8}$ | $0.01 \times 10^{-7}$ | $0.01 \times 10^{-7}$ |

**Table 11.** Results of the health impact (HI) of salt comparison of meal styles (Japanese and Western).

| HI of Salt Dalys Per Meal | Japanese (Model) | Japanese (Reduced-Salt) | Western (Model) | Western (Reduced-Salt) |
|---|---|---|---|---|
| Maximum data | $8.14 \times 10^{-6}$ | $4.60 \times 10^{-6}$ | $4.95 \times 10^{-6}$ | $4.70 \times 10^{-6}$ |
| Mean data | $5.21 \times 10^{-6}$ | $3.25 \times 10^{-6}$ | $3.40 \times 10^{-6}$ | $2.79 \times 10^{-6}$ |
| Median data | $4.70 \times 10^{-6}$ | $3.11 \times 10^{-6}$ | $3.10 \times 10^{-6}$ | $2.68 \times 10^{-6}$ |
| Minimum data | $2.92 \times 10^{-6}$ | $1.49 \times 10^{-6}$ | $1.46 \times 10^{-6}$ | $1.16 \times 10^{-6}$ |

### 3.2.2. Comparison of Meal Styles (Japanese and Western) on Environmental Impact Assessment

Table 12 shows the results of the environmental effects of climate change (CC) and water consumption (WC) for the ingredient stage of each meal style. Figure 3 shows the results of CC and WC for the ingredient stage of each dish. The CC value of the ingredient stage of the Japanese model meal was an average of $1.19 \times 10^{-6}$ DALYs per meal, in a range of $0.62 \times 10^{-6}$~$2.19 \times 10^{-6}$ DALYs per meal. The CC value of the ingredient stage of the Western model meal was an average of $1.78 \times 10^{-6}$ DALYs per meal, in a range of $0.71 \times 10^{-6}$~$5.43 \times 10^{-6}$ DALYs per meal, which was a seven-fold difference and 50% higher than the Japanese model meal value. The CC value was higher for Western meals than for Japanese meals, owing to a large amount of beef and other meats. The WC of the ingredient stage of the Japanese model meal was an average of $0.73 \times 10^{-6}$ DALYs per meal in a range of $0.19 \times 10^{-6}$~$1.03 \times 10^{-6}$ DALYs per meal, which was a ten-fold difference. The WC of the ingredient stage of the Western model meal was an average of $0.66 \times 10^{-6}$ DALYs per meal, in a range of $0.24 \times 10^{-6}$~$1.65 \times 10^{-6}$ DALYs per meal, which was a seven-fold difference. The average WC value was 10% higher for the ingredient stage of the Japanese meal than for the Western meal, owing to a larger amount of rice. To create a meal with low environmental impact (regarding climate change and water consumption), the ingredients should be changed from beef to pork, chicken or fish, and rice should be replaced with wheat noodles or bread.

This study considered a Japanese-style meal in which the main ingredients were salted dried fish, raw fish, and tofu, with the cooking methods of hot pot, simmering and fried rice. We also considered a Western-style meal comprised of sandwiches, gratin, spaghetti and curry, and the cooking methods were from overseas. This study categorized these meal styles for feature analysis. There was no clarifying definition of meal styles (Japanese and Western), and this study did not consider which meal style was better. We hope that all people will create sustainable meals by better understanding that the Japanese and Western food cultures have both good and problematic aspects.

**Table 12.** Results of the environmental effects of climate change (CC) and water consumption (WC) for ingredients of each meal style.

| Ingredient | Climate Change | | Water Consumption | |
|---|---|---|---|---|
| DALYs Per Meal | Japanese | Western | Japanese | Western |
| Maximum data | $2.19 \times 10^{-6}$ | $5.43 \times 10^{-6}$ | $1.03 \times 10^{-6}$ | $1.65 \times 10^{-6}$ |
| Mean data | $1.19 \times 10^{-6}$ | $1.78 \times 10^{-6}$ | $0.73 \times 10^{-6}$ | $0.66 \times 10^{-6}$ |
| Median data | $0.95 \times 10^{-6}$ | $1.24 \times 10^{-6}$ | $0.69 \times 10^{-6}$ | $0.46 \times 10^{-6}$ |
| Minimum data | $0.62 \times 10^{-6}$ | $0.71 \times 10^{-6}$ | $0.19 \times 10^{-6}$ | $0.24 \times 10^{-6}$ |

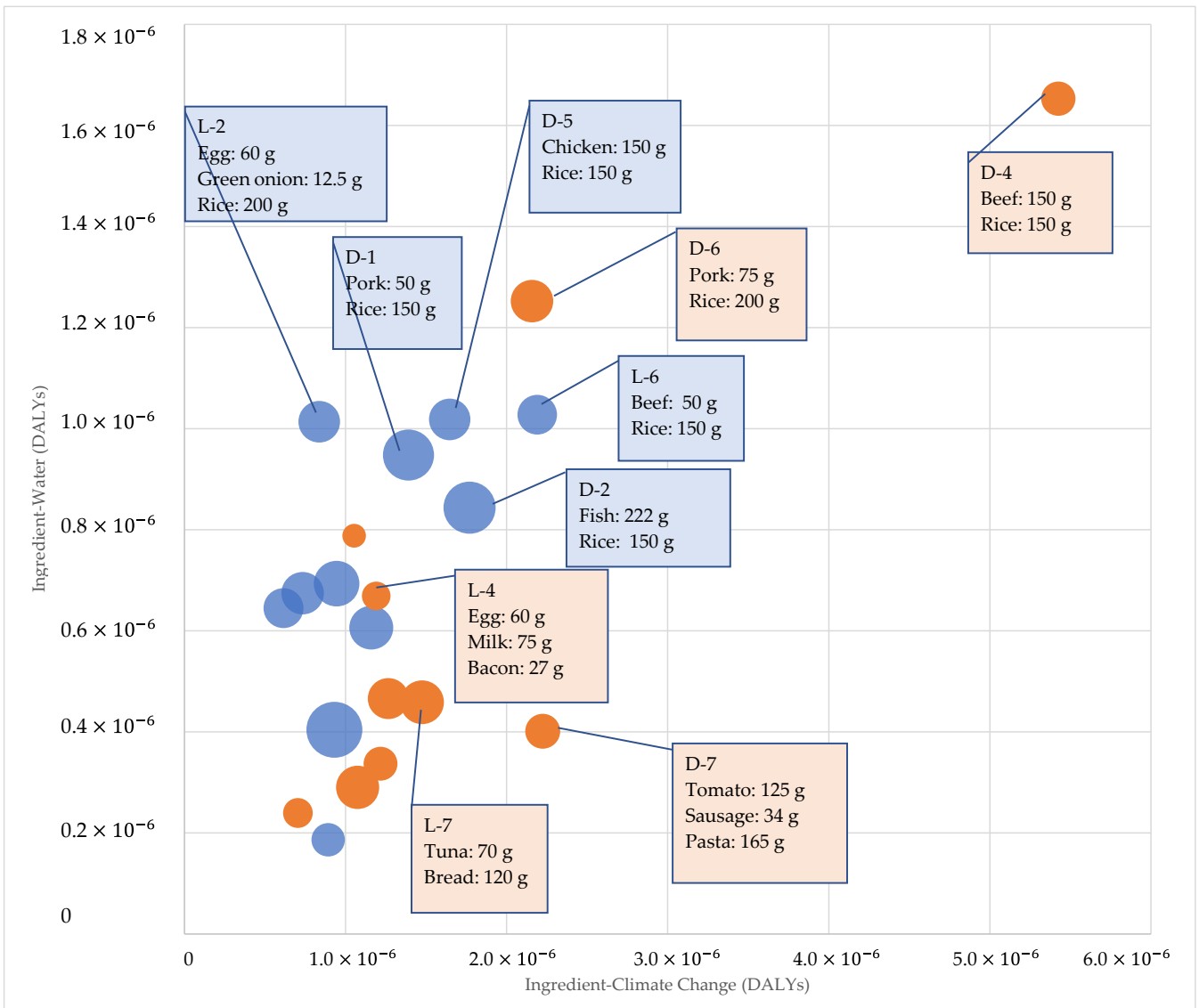

**Figure 3.** Results of climate change (CC) and water consumption (WC) for ingredients of the chosen meal.

### 3.2.3. Comparison of Breakfast, Lunch and Dinner

Tables 13 and A5 and Table A6 in Appendix A show the health impact, salt intake and calories for breakfast, lunch and dinner, respectively. Regarding the model breakfast, lunch and dinner, the calories, salt intake and health impact were 420, 582, and 806 kcal per meal; 2.6, 3.2 and 3.5 g per meal; and $5.4 \times 10^{-6}$, $6.4 \times 10^{-6}$ and $8.2 \times 10^{-6}$ DALYs per meal, respectively. All values for dinner were high, suggesting that changing the ingredients and reducing the salt in the dinner menu would benefit both the environment and health.

**Table 13.** Health impact of breakfast, lunch and dinner.

| Health Impact | Breakfast | | Lunch | | Dinner | |
|---|---|---|---|---|---|---|
| DALYs Per Meal | Model | Reduced-Salt | Model | Reduced-Salt | Model | Reduced-Salt |
| Maximum data | $7.14 \times 10^{-6}$ | $6.26 \times 10^{-6}$ | $9.51 \times 10^{-6}$ | $6.57 \times 10^{-6}$ | $10.2 \times 10^{-6}$ | $8.73 \times 10^{-6}$ |
| Mean data | $5.23 \times 10^{-6}$ | $4.50 \times 10^{-6}$ | $6.43 \times 10^{-6}$ | $4.86 \times 10^{-6}$ | $8.18 \times 10^{-6}$ | $6.57 \times 10^{-6}$ |
| Median data | $5.44 \times 10^{-6}$ | $4.56 \times 10^{-6}$ | $6.40 \times 10^{-6}$ | $4.74 \times 10^{-6}$ | $8.23 \times 10^{-6}$ | $6.35 \times 10^{-6}$ |
| Minimum data | $3.35 \times 10^{-6}$ | $3.05 \times 10^{-6}$ | $4.05 \times 10^{-6}$ | $2.63 \times 10^{-6}$ | $5.81 \times 10^{-6}$ | $4.81 \times 10^{-6}$ |

### 3.2.4. Verification with Previous Study

Table 14 shows a comparison of the effects of ingredients on climate change between this study and a previous study. The results in this study were multiplied by 52 (weeks). The results reported by Springmann et al. for all world regions (WLD), developed countries (DPD) and developing countries (DPG) were 1.27, 1.73 and 1.18 t $CO_2$e per year, respectively [10]. The results of this study appear reasonable because they are almost the same as those of Springmann et al. The value for red meat in this study was half that of the previous study, suggesting that Japanese people have a lower red meat intake than people in other developed countries. The result for cereals in this study was twice that of the previous study, suggesting that Japanese people have a higher cereal intake than people in other developed countries.

**Table 14.** Comparison of effects of ingredients on climate change between this study and a previous study. (t-$CO_2$e).

| | This Study | Springmann et al. | | |
|---|---|---|---|---|
| | | All World Regions | Developed Countries | Developing Countries |
| Red meat | 0.33 | 0.67 | 0.95 | 0.62 |
| Cereal | 0.25 | 0.15 | 0.09 | 0.16 |
| Fruits and veg | 0.06 | 0.07 | 0.11 | 0.07 |
| Dairy | 0.07 | 0.09 | 0.20 | 0.07 |
| Other | 0.42 | 0.29 | 0.38 | 0.25 |
| Total | 1.12 | 1.27 | 1.73 | 1.18 |

### 4. Conclusions

This study evaluated the impact of reduced salt intake on the environment (climate change and water consumption) and human health (DALYs), comparing a model meal based on the average food intake of Japanese people with a reduced-salt meal based on the model meal. We found a beneficial impact, which was that the reduced-salt meal would have a 20% lower impact on overall human health owing to a reduced incidence of salt-intake-related diseases despite the small increase in the health impact of reduced-salt seasonings.

This study conducted a health impact comparison of meal styles (Japanese and Western). The salt intake in the Japanese model meal had a 50% higher health impact compared to the Western model meal. The reason for this difference was that the Japanese meal contained dishes using high-salt seasonings, such as miso and soy sauce, and foods that used a lot of salt for curing, such as vegetables and seaweed, and fish boiled in soy sauce. The health impact of the high salt intake in the Japanese model meal can be reduced to nearly the same level as the Western model meal largely by using reduced-salt seasoning.

This study assessed the health impact on climate change and water consumption through a comparison of meal styles (Japanese and Western). The Western model meal had a 50% higher health impact on climate change than the Japanese model meal owing to the large amount of beef and other meats. The Japanese model meal had a 10% higher health impact on water consumption than the Western model meal, owing to the large amount of rice. To decrease the health impact on climate change and water consumption, ingredients

should be changed from beef to pork, chicken or fish, and from rice to wheat noodles or bread, in order to create sustainable meals.

Calories, salt intake and health impacts were assessed for the breakfast, lunch and dinner of the model meal. All aspects of dinner were high, suggesting that changing the ingredients and reducing salt at dinner would benefit both the environment and health.

**Author Contributions:** Conceptualization, K.N. and N.I.; methodology, K.N. and N.I.; validation, N.I.; formal analysis, K.N.; investigation, K.N.; resources, K.N.; data curation, K.N.; writing—original draft preparation, K.N.; writing—review and editing, N.I.; visualization, K.N.; supervision, K.N.; project administration, K.N. All authors have read and agreed to the published version of the manuscript.

**Funding:** This research received no external funding.

**Institutional Review Board Statement:** Not applicable.

**Informed Consent Statement:** Not applicable.

**Data Availability Statement:** Not applicable.

**Conflicts of Interest:** The authors declare that they have no conflicts of interest.

### Appendix A

**Table A1.** The calculation example (B-2 Rice omelette (half)).

| Ingredients | Input Amount (Grams) | $CO_2$ Intensity (kg-$CO_2$/kg) | Carbon Footprint | Water Consumption Intensity ($m^3$/kg) | Water Consumption Footprint |
|---|---|---|---|---|---|
| Steamed rice | 135 | 1.81 | 0.2444 | 0.96 | 0.1296 |
| Tomato | 15 | 1.33 | 0.0200 | 0.034 | 0.0005 |
| Green pepper | 7.8 | 2.07 | 0.0161 | 0.032 | 0.0002 |
| Onion | 12.5 | 0.33 | 0.0041 | 0.014 | 0.0002 |
| Chicken | 30 | 1.85 | 0.0555 | 0.24 | 0.0072 |
| Egg | 30 | 1.76 | 0.0528 | 0.23 | 0.0069 |
| Butter | 3.3 | 17.7 | 0.0584 | 1.39 | 0.0046 |
| Oil | 2 | 3.55 | 0.0071 | 0.97 | 0.0019 |
| Salt | 0.15 | 0.87 | 0.0001 | 0.0005 | $7.5 \times 10^{-8}$ |
| Ketchup | 15.2 | 4.09 | 0.0622 | 0.56 | 0.0085 |
| Pepper | 0.1 | 2.17 | 0.0002 | 0.043 | $4.3 \times 10^{-6}$ |
| Ingredient stage | 251 | Not applicable | 0.5209 | Not applicable | 0.1597 |
| Power (kWh) | 0.04 | 0.62 | 0.0248 | 0.00004 | $1.6 \times 10^{-6}$ |
| Gas ($m^3$) | 0.014 | 0.52 | 0.0073 | 0.00001 | $1.4 \times 10^{-7}$ |
| Production stage | Not applicable | Not applicable | 0.0321 | Not applicable | $1.7 \times 10^{-6}$ |

**Table A2.** Primary data obtained for input ratios for flavourings.

| Ingredients | Flavouring of Japanese Bouillon | Reduced-Salt Flavouring of Japanese Bouillon | Flavouring of Western or Chinese Bouillon | Reduced-Salt Flavouring of Western or Chinese Bouillon |
|---|---|---|---|---|
| Food-grade amino acids | 40% | 40% | 15% | 10% |
| Potassium salt | 0% | 10% | 5% | 10% |
| Aquatic food | 10% | 20% | 0% | 0% |
| Livestock food product | 0% | 0% | 20% | 20% |
| Salt | 30% | 0% | 40% | 20% |
| Sugar group | 20% | 30% | 20% | 40% |

**Table A3.** Climate change (CC), comparing breakfast, lunch and dinner.

| Climate Change | Breakfast | | Lunch | | Dinner | |
|---|---|---|---|---|---|---|
| kg $CO_2$ Per Meal | Model | Reduced-Salt | Model | Reduced-Salt | Model | Reduced-Salt |
| Maximum data | 1.14 | 1.15 | 149 | 1.50 | 3.68 | 3.68 |
| Mean data | 0.70 | 0.70 | 0.89 | 0.90 | 1.55 | 1.56 |
| Median data | 0.67 | 0.68 | 0.82 | 0.83 | 1.51 | 1.51 |
| Minimum data | 0.44 | 0.45 | 0.56 | 0.58 | 0.83 | 0.84 |

**Table A4.** Water consumption (WC), comparing breakfast, lunch and dinner.

| Water Consumption | Breakfast | | Lunch | | Dinner | |
|---|---|---|---|---|---|---|
| $m^3$ Per Meal | Model | Reduced-Salt | Model | Reduced-Salt | Model | Reduced-Salt |
| Maximum data | 0.17 | 0.17 | 0.22 | 0.22 | 0.36 | 0.36 |
| Mean data | 0.12 | 0.12 | 0.13 | 0.13 | 0.21 | 0.21 |
| Median data | 0.14 | 0.14 | 0.10 | 0.10 | 0.21 | 0.21 |
| Minimum data | 0.05 | 0.05 | 0.04 | 0.04 | 0.09 | 0.09 |

**Table A5.** Salt intake (SI), comparing breakfast, lunch and dinner.

| Salt Intake | Breakfast | | Lunch | | Dinner | |
|---|---|---|---|---|---|---|
| g Salt Per Meal | Model | Reduced-Salt | Model | Reduced-Salt | Model | Reduced-Salt |
| Maximum data | 3.9 | 3.2 | 5.8 | 3.2 | 5.1 | 3.4 |
| Mean data | 2.6 | 2.1 | 3.2 | 2.1 | 3.5 | 2.3 |
| Median data | 3.0 | 2.2 | 3.2 | 1.8 | 3.4 | 2.2 |
| Minimum data | 1.0 | 0.8 | 1.5 | 1.0 | 2.2 | 1.1 |

**Table A6.** Calories, comparing breakfast, lunch and dinner.

| Calories | Breakfast | | Lunch | | Dinner | |
|---|---|---|---|---|---|---|
| kcal Per Meal | Model | Reduced-Salt | Model | Reduced-Salt | Model | Reduced-Salt |
| Maximum data | 552 | 552 | 799 | 799 | 993 | 993 |
| Mean data | 420 | 420 | 582 | 582 | 806 | 806 |
| Median data | 416 | 416 | 551 | 551 | 760 | 760 |
| Minimum data | 312 | 312 | 370 | 370 | 676 | 676 |

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
