# Peer review of "Environmental and Health-Related Lifecycle Impact Assessment of Reduced-Salt Meals in Japan"

_sustainability, doi:10.3390/su14148265_

Round 1

Reviewer 1 Report

Nakamura and Itsubo presented a previously revised manuscript.

Through a study based on the Life Cycle Assessment method, the authors highlighted the environmental impact of two food scenarios: one based on traditional Japanese foods and one with reduced-salt meals.

The topic is very current due to the increasingly pressing need for an assessment of the environmental impact of human activities, including nutrition, and the health impact of high sodium consumption in both Western and Asian populations.

Although the study is still challenging to consult, the authors have significantly implemented the manuscript based on requests.

Author Response

Response to Reviewer 1:

We appreciate your careful review and valuable comments and suggestions. We have carefully revised the text according to your comments with references to the papers you kindly suggested.

  • Nakamura and Itsubo presented a previously revised manuscript. Through a study based on the Life Cycle Assessment method, the authors highlighted the environmental impact of two food scenarios: one based on traditional Japanese foods and one with reduced-salt meals. The topic is very current due to the increasingly pressing need for an assessment of the environmental impact of human activities, including nutrition, and the health impact of high sodium consumption in both Western and Asian populations. Although the study is still challenging to consult, the authors have significantly implemented the manuscript based on requests.

Thank you very much for your review comments. We have revised our manuscript following other reviewers’ comments.

Reviewer 2 Report

Dear Authors,

The manuscript (sustainability-1786878) submitted for review is interesting, but in my opinion, is not well written. I recommend it for publication after the authors' answers to the questions and after major revision

 Lines 151-152: I don't understand formulas 1 and 2. What does the data in the formula mean? This requires some explanation. Where did the authors get the data for substitution into the formula as (CO2 intensity)i,s? Where did the authors get the data for substitution into the formula as (Water consumption intensity)i,s? It is not explained in the manuscript.

Lines 174-175: The abbreviations for this formula should be explained. Clarifications are needed on how the calculations according to this formula were performed.

Lines 208-209 and lines 211-212: The method of counting according to these formulas also requires some explanation.

Figures 3,4,5,6,7,8,10- These Figures (box graphs) are unclear, perhaps it would be better to present these data in the form of a table,

Appendix: Tables A1 and A2, B2 and B1 - What is this data? There is no mention of this in the text of the manuscript.

Overall, while the article is interesting is not understandable. It would be better to explain how the calculations were made. Now, in my opinion, it is unclear and it will be difficult, for example, to make similar calculations for meals in another country. Other scientists need this study and article, but the method of study needs to be clarified,

Limitation

I have a question. Is there any limitation to these results? If yes, it is worth writing about it.

Conclusions

What are the practical and theoretical implications of this review? What are the authors' recommendations for scientists? The conclusions are too long. In this version, it is more of a summary

References:

References are cited according to journal rules. In my opinion, there are a lack of citation of articles related to this topic. The authors only cited reports.
 The manuscript needs to be improved.

Reviewer

Author Response

Response to Reviewer #2

We appreciate your careful review and valuable comments and suggestions. We have carefully revised the text according to your comments with references to the papers you kindly suggested.

  • Lines 151-152: I don't understand formulas 1 and 2. What does the data in the formula mean? This requires some explanation. Where did the authors get the data for substitution into the formula as (CO2 intensity)i,s? Where did the authors get the data for substitution into the formula as (Water consumption intensity)i,s? It is not explained in the manuscript.

We added following sentences to lines 152-155 in order to clear explain of intensity and from where its original data.

“We used greenhouse gas emission factor as CO2 intensity and water consumption factor as water intensity on IDEA v2.3. These factors were the item’s total CO2 equivalent emissions from cradle to gate, i.e., from nursery plants to cultivation using water and fertilizer on agricultural farms.”

  • Lines 174-175: The abbreviations for this formula should be explained. Clarifications are needed on how the calculations according to this formula were performed.

Formula 3 is revised to apply calculation method of the Discussion. We wrote explanation of the abbreviations of formula 3 in line 179.

Here, i is the item input and s is the lifecycle stage.

  • Lines 208-209 and lines 211-212: The method of counting according to these formulas also requires some explanation.

We added the detail explanation of formula 5 in lines 214-219.

“In concert with Formula 5 for the B-2 Rice omelette (half) in Table 1, the health impact of 1 gram of salt intake = 1.40 × 10-6 DALYs per gram was multiplied by the salt intake of B-2 = 1.0 gram, the difference between the impact of the salt model and reduced salt intake per meal in 0.2 grams was multiplied by the decreasing health impact index = 1.37 × 10-6 DALYs per gram; the health impact of the reduced-salt meal of B-2 = 1.126 × 10-6 DALYs was obtained by the former value minus the later value.”

We improved lines 221-224 as following.

“Regarding the one-week model meal and reduced-salt meal lists, the endpoint assessment results for DALYs at the ingredient and cooking stages were added to the health-impact DALYs for salt or reduced salt, and a comprehensive impact assessment was carried out:”

  • Figures 3,4,5,6,7,8,10- These Figures (box graphs) are unclear, perhaps it would be better to present these data in the form of a table. 

Thank you for your advice. We changed all box graphs from figures to tables.

  • Appendix: Tables A1 and A2, B2 and B1 - What is this data? There is no mention of this in the text of the manuscript.

We wrote that; table A1 in lines 156-157, table A2 in lines 158-159, table A3 (previous version figure B1) in lines 254-256 and table A4 (previous version figure B2) in lines 267-268.

  • Overall, while the article is interesting is not understandable. It would be better to explain how the calculations were made. Now, in my opinion, it is unclear and it will be difficult, for example, to make similar calculations for meals in another country. Other scientists need this study and article, but the method of study needs to be clarified.

Thank you for your advice. The text in formulas and explanation texts, are revised as above actions for easy understanding.

  • I have a question. Is there any limitation to these results? If yes, it is worth writing about it.

We wrote the limitation of the results in lines 317-328 as following, by improving and moving from previous version in lines 423-426.

“As a limitation, this study assessed the environmental and health-related effects of meals consumed by Japanese people with different levels of salt. The salt intake levels were created by replacing seasoning/salt with low-salt substitutes. In order to reduce the salt intake of Japanese people, it is important not only to replace ingredients with low-salt substitutes but also to reduce added salt, and educational interventions are very important.

This study covered only the data of 5743 Japanese people, without including other countries. The results evaluated average only intake, not that according to sex and age. This study assessed only salt intake, which is the most problematic nutrient for Japanese people. We hope that other nutrients, including the other 14 top GBD nutrients in order of priority, will be assessed comprehensively in terms of their environmental and health impacts by sex and age.”

  • What are the practical and theoretical implications of this review? What are the authors' recommendations for scientists? The conclusions are too long. In this version, it is more of a summary.

Thank you for your advice. We improved the conclusion by being more short sentences to easy understand the practical and theoretical implication for readers.

“This study evaluated the impact of reduced salt intake on the environment (climate change and water consumption) and human health (DALYs), comparing a model meal based on the average food intake of Japanese people with a reduced-salt meal based on the model meal. We found a beneficial impact that the reduced-salt meal would have a 20% lower impact on overall human health owing to a reduced incidence of salt-intake-related diseases, despite the small increase in the health impact of reduced-salt seasonings.

This study conducted a health impact comparison of meal styles (Japanese and Western). The salt intake in the Japanese model meal had a 50% higher health impact compared to the Western model meal. The reason for this difference was that the Japanese meal contained dishes using high-salt seasonings, such as miso and soy sauce, and foods that used a lot of salt for curing, such as vegetables and seaweed, and fish boiled in soy sauce. The health impact of the high salt intake in the Japanese model meal can be reduced to nearly the same level as the Western model meal largely by using reduced-salt seasoning.

This study assessed the health impact on climate change and water consumption through a comparison of meal styles (Japanese and Western). The Western model meal had a 50% higher health impact on climate change than the Japanese model meal owing to the large amount of beef and other meats. The Japanese model meal had a 10% higher health impact on water consumption than the Western model meal owing to the large amount of rice. To decrease the health impact on climate change and water consumption, ingredients should be changed from beef to pork, chicken or fish, and from rice to wheat noodles or bread, in order to create sustainable meals.

Calories, salt intake and health impacts were assessed for breakfast, lunch and dinner of the model meal. All aspects of dinner were high, suggesting that changing the ingredients and reducing salt at dinner would benefit both the environment and health.”

  • References are cited according to journal rules. In my opinion, there are a lack of citation of articles related to this topic. The authors only cited reports. The manuscript needs to be improved.

Thank you for your advice. We added citation of the Introduction in lines 70-72, that is Nutrients and environment comprehensive evaluation in US diet published by the Nature Food.

“Stylianou et al. [11] developed the Health Nutritional Index to quantify marginal health effects by combining nutritional health-based and 18 environmental indicators in the US diet, but did not evaluate in the Japan diet.”

Reviewer 3 Report

The manuscript was well improved and I congratulate the authors for that. There are some minor revisions before acceptance. The limitations were addressed in the conclusion section but should be moved to the end of the discussion section. The conclusion is still too long.

 Thank you for the opportunity to review this manuscript!

Author Response

Response to Reviewer #3

We appreciate your careful review and valuable comments and suggestions. We have carefully revised the text according to your comments with references to the papers you kindly suggested.

  • The manuscript was well improved and I congratulate the authors for that. There are some minor revisions before acceptance. The limitations were addressed in the conclusion section but should be moved to the end of the discussion section. The conclusion is still too long.

Thank you for your advice. We improved the conclusion by being more short sentences to easy understand the practical and theoretical implication for readers. We wrote the limitation of the results in lines 317-328, by improving and moving from previous version in lines 423-426.

Round 2

Reviewer 2 Report

Dear Authors,

The authors changed many parts of the planned paper according to my suggestions. They also added limitations to this study.

I would like to thank the authors for considering my comments and applaud them for the major revisions to improve their manuscript. In my opinion, the manuscript is now clear and more understandable than the previous version. Currently, I haven’t special comments, only technical comments. In lines 178-179, the authors still do not explain the shortcuts: CFP and WCFP.

I can recommend the manuscript for publication without correction.

Reviewer

This manuscript is a resubmission of an earlier submission. The following is a list of the peer review reports and author responses from that submission.

Round 1

Reviewer 1 Report

Nakamura and Itsubo proposed a study based on the LCA analysis of the typical Japanese meal, comparing it with the scenario of using low-sodium solutions (soy sauce, miso and other condiments subjected to desalination). The analysis of the environmental impact of the two scenarios through the evaluation of the effect on carbon and water consumption were compared with the beneficial effects on health derived from the reduction of salt in the diet. These aspects were also assessed with Western-style meal scenarios. The authors highlighted that despite a slight increase in the environmental impact that emerged in the low-sodium scenarios, derived from the influence of biotechnological processes for desalination, the impact on health has a positive effect.
The topic is central to public health policies as cardiovascular diseases continue to represent the leading cause of death and health care expenditure in the world. However, this is particularly relevant in the case of a Westernized diet, which is very rich in saturated fats and other favouring factors.
However, I have some doubts about some aspects:
- In the abstract, the nature and relevance of the effects of the low salt scenario on the environmental impact could be highlighted
- Being an LCA-based work, this could be highlighted in the abstract, moreover in the title
- On page 2, it is not clear why references 8 and 9 were mentioned. They don't seem to be relevant to the context
- The manuscript in the title, in the abstract and in general in its structure seems to be focused on Japanese meals. However, the analysis provides comparisons with Westernized nutrition. This should be better explained and properly motivated. It is not clear to me why, if the goal is to also compare Western and Japanese meals, why in the weekly model these are mixed and not separated into two scenarios, each weekly?
- It could be misleading to consider carbon as the sole climate impact. It may be clearer to speak in the text, figures and tables of carbon and water consumption, both phenomena that influence the environmental impact.
- Table 1 could be omitted and refer directly to supplementary materials
- I had a hard time finding correspondence between text and tables (for example on page 7 concerning table 3 and on pages 9-10 for table 5). I advise authors to recheck tables, specify the numerical values ​​to which they refer in the text, without using percentage approximations, and above all to declare if the differences are significant. It seems to me that the differences discussed are not statistically significant.
- Lines 271-273 specify that the impact of the reduced-salt meal is 30% lower. It's correct? In line 273 Authors say that this impact (again on single day analysis) is 20% lower. Am I missing something?
- The sentences on lines 301-303 and 304-306 seem ambiguous to me: 50% or 7-fold? 10-fold compared to what?
- On lines 346-351, is the impact on health 30 or 20% lower?
- In general, figures and tables should have more exhaustive captions describing the elements presented
- Discussion and conclusions should be better balanced. The discussion should comment on the results and discuss applicability. The conclusions are too extensive compared to the discussion (Table 6 seems to me to be irrelevant to the purpose of the study).
- The list of references should be standardized.

Reviewer 2 Report

Dear Authors,

The manuscript (sustainability-1736452) submitted for review is interesting, but not well written in my opinion. I recommend it for publication after the authors' answers to the questions and after major revision

 Lines 144-145: I don't understand formulas 1 and 2, what did the authors mean regarding the amount of input? This requires some explanation

Lines 166-167: Abbreviations for this formula should be explained.

Lines 196-197: The method of counting according to formula 4 should be explained

Figures 3,4,6,7, 9 - These Figures are unclear, perhaps it would be better to present these figures in the form of a table

Overall, while the article is interesting is not understandable.  It would be better to explain how the calculations were made. Now, in my opinion, it is unclear and it will be difficult, for example, to make similar calculations for a diet in another country. Other scientists need this study and article, but the method of study needs to be clarified

Limitation

I have a question. Is there any limitation to these results? If yes, it is worth writing about it.

Conclusions

What are the practical and theoretical implications of this review?

What are the authors' recommendations for scientists?

References:

References are not cited according to journal rules. Publications from MDPI provide information on how to properly cite. Authors may also find this information in the authors' guide.

 I have mixed feelings about recommending this manuscript for publication as it needs deep revision, but the idea for this manuscript is interesting.

Reviewer

Reviewer 3 Report

Dear Authors, Dear Editor,

In the manuscript reviewed, the authors presented the assumptions and results of a very interesting and valuable study. They looked at the recommendation to reduce salt intake through the lens of the impact of a diet altered in this direction not only on health, but also on climate change (carbon and water footprint). Such two-dimensional studies are essential as their results provide scientific evidence for the transformation of dietary patterns to healthy and sustainable ones.

The introduction provides information and data relevant to the research question. The material and methodology and the results are presented in detail and clearly. However, with regard to the discussion and conclusion, I have a substantive suggestion concerning the interpretation of the results in terms of dietary recommendations. In L. 333-337 you can write that the results are reasonable, but pointing to the same order of magnitude/level. But one cannot write that 0.33 is "almost the same" as 0.67 or 0.95 or 0.62. I would also suggest to check and write what is the average consumption of cereals and red meat in Japan, on average in the world and in the indicated categories of countries. These cannot be suggestions. In L. 366-368 the nutritional interpretation of the results is very superficial. If the recommendation for developed countries is to move towards a planetary diet, then it cannot be suggested to replace beef with other meats or fish. The aggregate consumption of this high-protein animal food in Japan is very high, similar to the average in EU countries (according to FAOSTAT, Food Balance Sheets, is close to 100 kg/person/year). Also, I question the suggestion of swapping rice for wheat pasta or bread, with no indication for whole grain products.

Minor comments:

L.20. I suggest adding that it is a negative impact, and in L.24 - a beneficial one. I also suggest introducing these unambiguous terms in the conclusion (L.340-350).

L.42 and 47. In CO2 notation, 2 can be written in subscript (as in the other 4 cases). 

L.43-44. This sentence is an understatement and may be misleading. The 57% share refers to GHGs emissions from global food systems; I suggest clarification: The livestock industry accounts for 57% of GHGs emissions in the world's food systems and food agriculture for 29%; the remaining 14% comes from other agricultural products such as cotton and rubber.

L.106. 130 kcal times 7 is 910 kcal and this value could have been used for calculations (but they were already done, surely taking 5 kcal more did not determine the results).

Table 1. Instead of this table I suggest inserting Table A1 here - substantively they do not differ much and it will be better to see the full sets of meals at once. Moreover, you can reduce the leading and resign from writing 'styles' (3 lines less), the table will be clearer.

Table 3 and 5. Under the tables, please include an explanation of the abbreviation ND.

Figure 8. Instead of “each” I suggest 'chosen'.

Section 4.1. and Table 6. Abbreviations: WLD, DPD, DPG are redundant. In the title of the table it is useful to write the unit.

L.336 and 337. Instead of “other” I suggest 'developed'.

And finally, I still have a question about Figures B1 and B2. Could the fact that for breakfasts there were 3 Japanese style and 4 Western style, and for lunches and dinners there were 4 Japanese and 3 Western style determine the differences in environmental impact?

Kind regards.

Reviewer 4 Report

The manuscript “evaluated the impact of reducing salt intake on the environment and human health, comparing a model meal based on Japanese people’s average food intake with a reduced-salt meal by replacing seasoning/salt with low-salt substitutes based on the model meal”. The theme is very interesting, but it is necessary for some reviews that I highlight below.

-       Lines 12 – 14: It is hard to read the sentence. You should separate your objective (up to “human health”) from the method used (comparing a model….).

-       Abstract: the method and sample size are not clear.

-       Line 36: sodium is different from salt. Please correct it and explain it better.

-       Line 37: explain “lifestyle disease”

-       Line 38: consumption?

-       Lines 39-41: also mention the total amount of sodium since you mention sodium as the main cause.

-       Lines 42-49: Do you have data on GHG emission in salt extraction/production/commercialization?

-       Lines 52-54: removing salt or sodium?

-       The objective should be more clear: are you evaluating salt or sodium reduction? Is it from replacing salt with reduced-salt seasonings or only reducing foods with high amounts of sodium? Highlight the research gap and your objectives. It could be interesting to use the hypothesis or research question.

-       Lines 74 – 79: methods.

-       Lines 82-83: did you consider the same amount? It is important to verify that the use of different products affects the sensory quality. Many times it is necessary to adjust the quantity of salt substitutes in the product. Also, it is frequent to use spices and herbs to substitute salt. They are healthier than industrialized reduced-sodium seasonings. Did you consider it? If it is a habit in your country, please, highlight it in the introduction section. In my country, sodium reduction s performed using spices and herbs.

-       Line 102: wheat-bread.

-       Lines 113-123: Only here I can understand the objective. The introduction and objectives should highlight this.

-       It should be more clear that the process of low-sodium seasonings is the main focus of evaluating GHG.

-       In food production, the reduction of sodium might impact cooking time and process. Did you consider it?

-       Did you consider the health impact of low-sodium products? It is known that they have less sodium but can also impair health. Include the information on the specific health impact you are evaluating.

-       Table 2: Can low-sodium products impact these diseases?

-       Results and discussion were placed together. It should be better to separate it.

Thank you for the opportunity to review this manuscript!

Reviewer 5 Report

Dear Authors:

I like the work you have done but there are some things that you need to improve to make this paper a much better one suitable for publication: These are:

1) Explain what ND means in Table 3. It's somewhat confusing because it is under the PValue column. If it means the PValue was not determined, provide the reasons why it was for other results and not for those in rows of which the PValue is ND. Please correct this confusion.

2) Provide three bold lines in Tables, not more than 3 as it is the standard for academic journal articles.

3)Remove the lines within the Figure 3. Also provide the sub-title for the x-axis (Styles). The current way the figure is looks messy. Doing what I have suggested will improve the quality of your work.

4) Remove the lines within the Figure 4. Also provide the sub-title for the x-axis (Styles). The current way the figure is looks messy. Doing what I have suggested will improve the quality of your work.

5)Remove the lines within the Figure 5. Also provide the sub-title for the x-axis. The current way the figure is looks messy. Doing what I have suggested will improve the quality of your work.

6) Remove the lines within the Figure 6. Also provide the sub-title for the x-axis (Meal Styles). The current way the figure is looks messy. Doing what I have suggested will improve the quality of your work.

7) Remove the lines within the Figure 7. Also provide the sub-title for the x-axis. The current way the figure is looks messy. Doing what I have suggested will improve the quality of your work.

8)Remove the lines within the Figure 8. The current way the figure is looks messy. Doing what I have suggested will improve the quality of your work.

9)Remove the lines within the Figure 9. Also provide the sub-title for the x-axis because one would have tp search the label of what is on the x-axis. The current way the figure is looks messy. Doing what I have suggested will improve the quality of your work.

10) The lines in Table A1 make the table look messy, only three bolded lines are required and not lines all over. Please correct this.

11)Remove the lines within all the Figures in Appendix. Also provide the sub-title for the x-axis because one would have to search the label of what is on the x-axis. The current way the figure is, looks messy. Doing what I have suggested will improve the quality of your work.

Thank you!